# CD4 T Cell-Mediated Immune Control of Cytomegalovirus Infection in Murine Salivary Glands

**DOI:** 10.3390/pathogens10121531

**Published:** 2021-11-23

**Authors:** Nathan Zangger, Josua Oderbolz, Annette Oxenius

**Affiliations:** Institute of Microbiology, ETH Zurich, 8093 Zurich, Switzerland; nzangger@biol.ethz.ch (N.Z.); josua.oderbolz@gmail.com (J.O.)

**Keywords:** CD4 T cells, cytomegalovirus infection, salivary gland

## Abstract

CD4 T cells are well known for their supportive role in CD8 T cell and B cell responses during viral infection. However, during murine cytomegalovirus (MCMV) infection in the salivary glands (SGs), CD4 T cells exhibit direct antiviral effector functions to control the infection. In this mucosal organ, opposed to other infected tissues, MCMV establishes a sustained lytic replication that lasts for several weeks. While the protective function of CD4 T cells is exerted through the production of the pro-inflammatory cytokines interferon gamma (IFNγ) and tumor necrosis factor alpha (TNF), the reasons for their markedly delayed control of lytic MCMV infection remain elusive. Here, we review the current knowledge on the dynamics and mechanisms of the CD4 T cell-mediated control of MCMV-infected SGs, including their localization in the SG in relation to MCMV infected cells and other immune cells, their mode of action, and their regulation.

## 1. Introduction

Cytomegalovirus (CMV) and human herpesvirus 5 (HHV-5) are large double-stranded DNA viruses of the β-herpesvirus family. Like all herpesviruses, they establish a lifelong persistence in the form of latency in susceptible hosts after control of the primary lytic infection. Seroprevalence is high across human populations, ranging from 50% to 90%, reaching high epidemiological relevance [1,2,3]. Although primary infection in immunocompetent hosts usually proceeds asymptomatically, causing no severe complications, immunocompromised people such as transplant recipients or HIV patients can suffer from CMV infection with serious clinical manifestations like end organ disease [4,5,6]. In addition, in utero CMV transmission to the fetus during pregnancy can have severe consequences for the newborn, such as hearing loss, developmental and motor delay, vision loss, and microcephaly [7]. Finally, CMV infection has also been suggested to be associated with cardiovascular disease and increased immune senescence, an age-associated impairment of the immune system [8,9,10].

Sharing over 80% of genome similarity, murine CMV (MCMV) has been used to better understand the CMV life cycle as well as the immune control of and immune evasion by CMV [11,12]. Due to its broad tropism, CMV infection occurs in various tissues and proceeds in three distinct phases: lytic, persistent, and latent. First, a systemic and lytic replication phase takes place in many organs, such as the spleen and the lungs. Thus, MCMV primary infection induces a robust, innate, and adaptive immune response, leading to the control of viral replication in most organs within roughly one to two weeks [13,14] (Figure 1A). Of note, the viral kinetics mentioned here are based on the infection with the genetically engineered virus which expresses MCK-2 in addition to Gaussia luciferase, mCherry, and an additional sequence within the m164 ORF encoding the SIINFEKL peptide. Subsequently, a persistent phase may result from the acute infection due to diverse immune evasion mechanisms, in particular in the salivary glands (SGs). These are a preferred mucosal site of persistent CMV replication, playing a central role in horizontal transmission in both humans and mice. In the SG, lytic replication continues to increase until four weeks post-infection, and is controlled only about two months after infection [15,16] (Figure 1B). However, despite an effective resolution of lytic viral replication, the immune system fails to completely clear the virus as it enters latency. This phase is characterized by the interruption of the lytic transcription program by epigenetic modifications, leading to the suspension of the production of infectious virions [17] (Figure 1C). Importantly, this latent transcription program can be reversed to a lytic transcription program by diverse factors, such as inflammation and oxidative stress. While reactivation events occur sporadically throughout the life of the immunocompetent host, they are rapidly controlled by potent CMV-specific T-cell responses [18,19]. 

Adaptive immunity is a pivotal component of CMV control, restricting primary infection, and preventing reactivation, which ensures a largely benign host-virus equilibrium. Both CD4 and CD8 T cells play an important role in controlling MCMV infection. Although cytotoxic CD8 T cells can control lytic infection in most organs, and despite a significant higher ratio in the SGs during MCMV infection, early studies defined this mucosal organ as an exception, where only virus-specific CD4 T cells are able to achieve control of lytic replication [15,20,21,22]. Among them, IFNγ producing CD4 T cells—reflecting a Th1 response—are the main drivers of protection and the main type of response in this organ. Nevertheless, the MCMV-specific CD4 T cell immune response exhibits a wide repertoire with specificities for various antigenic peptides and, interestingly, CD4 T cell populations being specific for distinct peptide antigens follow different kinetics [23].

In spite of an early infiltration of MCMV-specific CD4 T cells into the SGs, control of lytic MCMV infection is slow and takes several weeks, whereas it is rapidly controlled in the spleen or the lungs [24]. One proposed explanation for this delayed control in the SGs has been the production of the immunosuppressive cytokine IL-10 by MCMV-specific CD4 T cells [25]. However, the exact mechanisms that are responsible for this delayed control remain to be further elucidated. 

In this review, we summarize the current understanding of the immune control mediated by CD4 T cells in MCMV-infected SGs. We describe the early events of CD4 T cell infiltration, and by which molecular mechanisms they position themselves in the SG tissue. Next, we outline the nature and kinetics of the general and MCMV-specific CD4 T cell response, along with their mode of action and regulation. Finally, we describe the control exerted by CD4 T cells during latency to prevent reactivation and re-infection.

## 2. Chemokine-Driven CD4 T Cell Dynamics in the SGs

Upon MCMV infection, effector CD4 T cells infiltrate the SGs after approximatively one week of infection, where they form discrete clusters. Most infiltrated CD4 T cells are then CD44^hi^ LFA-1^+^CD49d^+^, indicative of an effector-like phenotype [26,27,28]. Their numbers accumulate during the second week, reaching their peak, before contracting thereafter. However, significant numbers persist along the ongoing MCMV infection [23,24,25], and from four weeks onwards, a majority of MCMV-specific T cells display a memory signature [29].

As the CD4 response in the SGs has been shown to be mainly of a Th1 phenotype, several studies investigated the role of the signature markers CXCR3 and CCR5 as possible driving factors of SG infiltration [30,31]. The chemokine receptor CXC motif 3 (CXCR3) is an important chemokine for T cell migration. It binds to three IFNγ-inducible chemokines, namely CXCL9 (MIG), CXCL10 (IP-10), and CXCL11 (I-TAC), with the highest affinity for CXCL11 and the lowest for CXCL9 [32,33,34]. The C-C chemokine receptor type 5 (CCR5) binds to CCL3 (MIP-1α), CCL4 (MIP-1β), and CCL5 (RANTES) [35]. Although the splenic MCMV-specific CD4 T cells express Th1-associated chemokine receptors at the peak of the response (8 dpi), the SG-resident MCMV-specific CD4 T cells express barely detectable levels of CXCR3 at 8 dpi, and only low levels of CXCR3 and CCR5 at 14 dpi in total CD4 T cells [24,36]. This may be explained by a rapid internalization of these receptors following binding to their chemokine ligands within infected tissue. Indeed, a high expression of the respective chemokines is reported in MCMV-infected SGs [22]. 

However, recent data from Oderbolz et al. showed that CXCR3 expression is not necessary for T cell infiltration into the SGs [24]. Interestingly, in the case of CD8 T cells, recruitment to uninfected SGs is mediated by CXCR3 and the integrin α4, but redundant mechanisms mediate T cell recruitment to the SGs of MCMV-infected mice [37]. However, in the case of the liver, CD8 T cell recruitment is CXCR3-dependent during acute MCMV infection [38]. Even though precise data on CD4 T cell entry into the MCMV-infected SGs are missing, the redundancy or necessity for CXCR3 expression on MCMV-specific T cells for their recruitment during MCMV infection might be organ-dependent. 

At this point, no data are available on the migration behavior of MCMV-specific CD4 T cells after immigration to the SG. However, data for CD8 T cells have shown that they crawl along macrophage networks within the SG [39].

The micro-anatomical localization of CD4 and CD8 T cells in the SG during acute MCMV infection is reported to be quite distinct. Despite a higher abundance compared to CD4 T cells, CD8 T cells show a less marked clustering tendency, whereas CD4 T cells form discrete clusters around myeloid cells that harbor engulfed remnants of MCMV-infected cells [24]. Although CXCR3 does not seem to be required for the SG entry of MCMV-specific CD4 T cells, recent data from our lab showed a clear role for CXCR3 in CD4 T cell cluster formation, where CXCR3-competent CD4 T cells positioned themselves preferentially at CXCL9/10 hotspots [24]. Even if the exact cellular source of CXCL9/10 during MCMV infection remains to be determined, several studies on similar phenotypes offer some cues for speculation. It is known that salivary gland epithelial cells (SGECs) can secrete CXCL9-11, and that a moderate expression of CXCR3 on ductal epithelial, and occasionally on acinar epithelial cells, can be observed [40]. There, CXCR3 expression plays a scavenging role of chemokines to avoid local accumulations at a steady state. A defect of this mechanism, as seen in Sjögren’s syndrome, could be a reason for the accumulation of CD4 T cells [41]. Macrophages are also suggested to be a source of CXCL9/10 for the formation of CD8 tissue resident memory T cells (T_RM_) clusters after MCMV infection [39]. Therefore, it is conceivable that CD4 T cells are attracted by CXCL9/10 sources, being myeloid cells or SGECs in the vicinity of sites of infection, in order to facilitate the encounter of MCMV antigen-presenting cells and consequently to perform their effector functions, such as IFNγ production. Elucidating the precise cellular sources of CXCL9/10 by in situ studies might further emphasize their orchestrating role within these immune hubs.

Interestingly, besides the Th1-related chemokine receptor CXCR3, other chemokine receptors were also found on SG-resident CD4 T cells, in particular during later stages of infection. Once viral infection is controlled, CD4 T_RM_ express CCR5 and are found around the epithelial layer in human oral mucosal tissues [42]. Furthermore, a report demonstrating the presence of the chemokine CCL22 in both HCMV infection and Sjögren’s syndrome could potentially indicate a role for the CCR4-CCL22 axis [43,44]. As CCR4 is preferentially expressed by Th2-polarized CD4 T cells, the recruitment of a less effective Th response against viral infections could support immune evasion. 

Thus far, currently available studies point towards an organ-dependent role of CXCR3 and CCR5 for the organ infiltration of T cells, and in the case of the MCMV-infected SGs, both chemokine receptors seem to be redundant. However, once infiltrated into the SG, CD4 T cells are attracted to sites of infection by CXCL9/10 gradients, produced by either myeloid or epithelial cells of the SG. A role for other chemokines receptors is not excluded, and similar mechanisms might be necessary for the positioning of CD4 memory T cells at later stages. 

## 3. Phenotype of the MCMV-Specific CD4 T Cell Response in the SGs 

Early experiments by Jonjic et al. demonstrated that the control of MCMV infection in different organs such as the lungs or the spleen can proceed with comparable kinetics if either the CD4 or the CD8 T cell population is missing, suggesting a redundancy in T cell-mediated protection. This work also highlighted the SGs as an exception, where CD4 T cells were absolutely necessary for MCMV control, as the SGs responded drastically to a selective CD4 T cell depletion, exhibiting elevated virus titers [20,21]. Indeed, MCMV possesses a remarkable repertoire of immune evasion genes, especially to interfere with the MHCI presentation pathway in infected cells, which might explain the inability of MCMV-specific CD8 T cells to control MCMV infection in the SG [45]. The inability of MCMV-specific CD8 T cells to recognize MCMV-derived antigens in actively infected cells, combined with the inability of SG-resident APCs to cross-present MCMV-derived antigens [45], confers a central role to CD4 T cells in viral control in the SG.

As aforementioned, the earliest reported time of CD4 T cell infiltration into the SG is after one week, and the MCMV-specific CD4 T cell response is composed of cells with different antigen specificities. Nonetheless, for most targeted epitopes, the MCMV-specific CD4 T cell response follows similar kinetics (Figure 2). Of note, MCMV genes with the uppercase prefix “M” share homologs in the HCMV genome, whereas genes with the lowercase prefix “m” share no sequence identity with HCMV genes.

Based on the work of Arens et al., the numbers of MCMV-specific CD4 T cells were estimated to be ~6% of the total population of CD4 T cells at 8 dpi, with 14 epitopes triggering ~25% of the total MCMV-specific CD4 T cell response, based on antigen-induced IFNγ production [23]. Of note, this study was performed on splenocytes, and PMA/ionomycin was used to estimate the total MCMV-specific CD4 T cell response. In human studies, CMV-specific CD4 T cells in PBMCs, measured by antigen-induced IFNγ production, ranged from 0.42% to 2.5% [46]. Despite the focus on the IFNγ^+^ CD4 T cell response, the MCMV-specific CD4 T cell responses were shown to be diverse and polyfunctional, producing multiple cytokines (IFNγ, TNF, IL-2, IL-10 and IL-17) [23,36,47]. Therefore, these numbers could turn out to be higher when including a wider array of cytokines. 

Recent data from Oderbolz et al. employed the adoptive transfer of M25-specific TCR transgenic CD4 T cells prior to MCMV infection. Using this approach, the highest percentage of M25-specific CD4 T cells was observed at 8 dpi, with 10% in the spleen, almost 20% in the lung, and almost 30% in the SG [24]. The kinetics of endogenous MCMV-specific CD4 T cell responses was analyzed in the SG, focusing on the following epitope specificities: m09_133–147_, M25_409–423_, M25_411–425_, M78_417–431_, m139_560–574_, and m142_24–38_. We summarize the current knowledge of MCMV-specific CD4 T cell populations, recognizing specific epitopes in the SG in Table 1.

It is generally accepted that MCMV infection mainly induces a Th1 response. Consistent with this notion, Lucin et al. showed the pro-inflammatory cytokine IFNγ being a central component of the antiviral response against MCMV. Interestingly, whereas systemic IFNγ neutralization heavily inhibited MCMV-directed effector functions and caused an elevated viral load in the SGs, it barely affected the CD8 T cell antiviral activities in the spleen and the lungs. However, control of MCMV replication in the SGs could not be rescued by the systemic administration of recombinant IFNγ, suggesting that IFNγ alone cannot explain the full complexity of the CD4 T cell-mediated response [48]. Nevertheless, it is conceivable that the recombinant IFNγ might have not reached the SG in sufficient concentrations, as mixed bone marrow chimera experiments seemed to challenge the above interpretation. Indeed, a clear lack of MCMV control was observable in the SG in mice with CD4 T cells deficient for IFNγ production [45]. Therefore, it was hypothesized that IFNγ might exert both direct and supportive functions, such as directly inhibiting MCMV replication but also inducing downstream signaling targets on IFNγ-sensing cells (e.g., MHC class I and II molecules).

Selective deficiency of IFNγ production by CD4 T cells resulted in the severely compromised control of MCMV replication in the SGs [45]. Moreover, in vivo activated M25-specific CD4 T cells, which recognize the I-A^b^-restricted immunodominant epitope M25_411–425_ [23,47], when adoptively transferred into immunocompromised hosts, showed strong protection capacities during the acute phase of MCMV infection. This is demonstrated by significantly reduced viral loads in various organs, including the SGs, where the extent of virus control correlated with the number of transferred M25-specific CD4 T cells. These studies showed that, beyond having just a supportive and organizational role, IFNγ secretion by CD4 T cells provided direct antiviral effector functions, as IFNγR^−/−^ mice, reconstituted with M25 CD4 T cells, were unable to control the infection. In summary, these data emphasize a role of IFNγ-secreting MCMV-specific CD4 T cells in combating MCMV infection, particularly in the SGs. 

Several studies confirmed the central role of the Th1 response in MCMV infection, but also acknowledged the presence of additional cytokines. The work of Walton et al. [47] showed that the MCMV-specific CD4 T cell response in the lungs was dominated by Th1 lineage cytokines, as seen by the high frequencies of CD4 T cells exhibiting TNFα and IFNγ double production, but no IL-4 or IL-17-secreting CD4 T cells. IL-2-secreting CD4 T cells were only observable at later time points of infection, and very low frequencies of IL-10-secreting CD4 T cells were detected in this organ [47]. In addition, the work of Arens et al. describes CD4 T cells that mostly produce IFNγ and TNF together, however, CD4 T cells also producing IL-17, IL-2 and IL-10 were observed in the spleen. Of note, the production of IL-17 and IFNγ was mutually exclusive within epitope-specific CD4 T cells [23]. Such an in-depth analysis of the heterogeneity of the MCMV-specific CD4 T cells’ response in the SG is missing, and organ-specificity might presumably be present. 

CD4 T cells can use different mechanisms to mediate antiviral defense. Besides the production of antiviral cytokines, both mouse and human studies reported that perforin and granzymes are the main mediators of cytolytic activity in CD4 T cells. However, TNF ligands (i.e., FasL and TRAIL) could also play a role [27]. In the spleen, granzyme-expressing M78-specific CD4 T cells were shown to kill MCMV-infected target cells in an epitope-specific manner. Interestingly, this cytotoxic activity was organ specific, as M78-specific CD4 T cells in the SGs only showed an activated phenotype (CD44^+^ CD69^+^ CD43^+^ KLRG1^+^), but no cytotoxic activity. Supporting this notion, mice in which CD4 T cells lacked perforin controlled MCMV replication similar to mice with fully functional CD4 T cells [45]. Even though other mediators of CD4 T cell cytolytic activity have not been yet investigated, the absence of evidence of a role of MHC class II expression on infected epithelial cells for the control of MCMV replication in the SG (unpublished data) would suggest the absence or negligible role of cytotoxic CD4 T cells in the control of lytic MCMV replication in the SG. 

In addition to IFNγ production, IL-10-producing CD4 T cells were observed during MCMV infection in different organs [23,26]. Possibly representing a counterbalance to pro-inflammatory cytokines, a significant rise and maintenance of IL-10-producing CD4 cells was reported with different kinetics across organs. Whereas the frequencies of splenic IL-10 producing CD4 T cells contract rapidly after a peak at 5 dpi, their counterpart in the SGs displayed delayed kinetics, peaking at 14–30 dpi on both the RNA [22,25] and protein expression level with 4–12% of IL-10 producing CD4 T cells of total CD4 T cells, depending on the use of IL-10 reporter mice or ICS [36,49]. The generation of IL-10-producing CD4 T cells during MCMV infection seems to be dependent on differential signals in distinct organs. IL-27 signaling is proposed to induce the production of IL-10 in the spleen [50], whereas ICOS signaling is suggested to be the cause of IL-10 production in CD4 T cells in the SG [36]. Interestingly, both SGECs and APCs can express ICOS-L [51,52], but the exact cellular source of ICOS-L is yet to be determined in the MCMV-infected SGs. Of note, the cellular source of IL-27 during MCMV infection is attributed to dendritic cells (DC), and DC-derived IL-27 has been shown to restrict a cytotoxic program in MCMV-specific CD4 T cells, including the expression of the cytotoxic effector molecule granzyme A [50]. 

Although IL-10 is a signature cytokine of FoxP3^+^ CD4 regulatory T cells (Tregs), this population does not account for the vast majority of IL-10-producing CD4 T cells in the MCMV-infected SGs, the lung, and the spleen. CD4 FoxP3^+^ Treg levels are reported to decrease upon acute infection in the spleen and lungs before coming back to normal levels at later time points [53], with higher numbers in the BALB/c background [54]. However, they showed a more activated phenotype at early time points in the spleen, lungs, and SG [53]. On the contrary, levels of IL-10 producing FoxP3^+^ Tregs are increased and maintained during MCMV latency [16]. During latency, these Treg cells prevent IL-10 production from FoxP3^−^ CD4 T cells, and hence support the restriction of viral replication in the SGs, suggesting a similar function during acute infection. 

In summary, we highlight that MCMV-specific CD4 T cell responses are dominated by a Th1 lineage in the MCMV-infected SG. Studies in other organs revealed heterogeneity of MCMV-specific CD4 T cells, in terms of antigen-specificity and cytokine profiles, which might follow different kinetics. In the SGs, a significant proportion of MCMV-specific CD4 T cells produces IL-10 in addition to conventional Th1 cytokines, possibly arising from ICOS signaling, and this IL-10 production might contribute to the delay in viral control. 

## 4. Mode of Action and Regulation of MCMV-Specific CD4 T Cells during Acute Infection 

As the Th1 response is a central element of the immune-mediated control of MCMV infection in the SG, we will mainly focus here on describing the mode of action of IFNγ-producing CD4 T cells and their regulation. We will also briefly mention less investigated modes of action.

In the MCMV-infected SG, CD4 T cells are the major cellular source of IFNγ production [24]. However, it was shown that the antiviral effects of CD4 T cells require also the presence of TNF in the lung and the SG [55]. Accordingly, significant frequencies of co-producing MCMV-specific CD4 T cells were reported in the lung [47], and a substantial proportion of M25-specific CD4 T cells were shown to express IFNγ, but surprisingly low frequencies of IFNγ and TNF co-producing were detected in the SG. This does not exclude a higher frequency of IFNγ-TNF co-production in other MCMV-specific CD4 T cells.

In the SG, IFNγ-producing CD4 T cells tend to accumulate in high abundance within clusters. The presence of Nur77^+^ cells (Nur77 being a reporter of TCR signaling [56]) within these clusters suggests local antigen recognition. Interestingly, these clusters often contain CD11c^+^ myeloid cells that exhibit intracellular compartments that have acquired remnants of MCMV-infected cells (e.g., apoptotic bodies of previously infected cells) [24]. These MCMV-specific CD4 T cells were shown to also produce IFNγ as a consequence of TCR stimulation [24].

Based on these and additional data, a mathematical model was developed that simulates the interaction between CD4 T cells and MCMV antigen-presenting cells in relation to MCMV control in the SG. This model proposes a locally confined protective capacity of clusters of activated MCMV-specific IFNγ producing CD4 T cells rather than a long-range protection across the entire SG tissue. It further suggests that originating from the clusters of activated MCMV-specific CD4 T cells, an IFNγ concentration gradient is established that demarcates tissue areas that are protected from MCMV replication. Importantly, the IFNγ production by MCMV-specific CD4 T cells and the ensuing antiviral protection occurs in a delayed manner due to the indirect antigen recognition by effector CD4 T cells on CD11c^+^ APCs following the engulfment of cargo from previously infected cells. This secondary, indirect, and delayed recognition of viral infection could allow MCMV to spread further and replicate at tissue sites still unexposed to IFNγ [24]. This hypothesis would support the idea that a short-range diffusion of IFNγ, defined as a confined action radius of about 80 µm away from the IFNγ producing CD4 T cell clusters, could induce potent protection in a confined area around the interaction sites of MCMV-specific CD4 T cells and the MCMV-derived antigen presenting cells [57]. This indirect antigen recognition model could potentially explain the persistent ongoing viral replication over several weeks in SGs, where ultimately, a control of an MCMV lytic infection would only be reached once the continued accumulation of locally IFNγ-protected areas would result in an organ-wide protection. 

Besides these beneficiary effects of local IFNγ production, there needs to be additional measures to regulate such a response, as otherwise, it might induce immunopathology. Consistent with this notion, Schuster et al. showed that NK cells in the SG specifically eliminate activated CD4 T cells during persistent MCMV infection in a TNF-related apoptosis-inducing ligand-dependent (TRAIL) manner. In the absence of this regulation, persistent CMV infection was associated with a Sjögren-like syndrome, highlighting NK cells as an important regulatory cell population during MCMV infection [58]. Of note, the origin of the accumulating population of TRAIL^+^ NK cells in the SG described in the above work during MCMV infection was not determined. However, additional work highlighted the SG-resident ILC1 population as the major cell type to produce TRAIL during MCMV infection, while conventional NK express almost none. This would suggest a regulation mechanism of activated CD4 T cells by TRAIL^hi^ ILC1s [59]. 

Together, these studies suggest that CD4 IFNγ-producing T cells locally protect the MCMV-infected SG by clustering around APCs with engulfed viral remnants and through TCR triggering; this secreted IFNγ acts within a short range of the producing cells. Organ-wide protection would be reached ultimately once enough local areas are protected. To prevent immunopathology, TRAIL^+^ ILC1s are suggested to regulate the CD4-mediated IFNγ response. 

## 5. Late CD4 T Cell Responses during MCMV Infection

Interestingly, the CD4 T cell-mediated immune control of MCMV in the SGs is not only restricted to time points when viral replication ceases, but continues to contribute to protection against re-infection and reactivation at later stages. Thom et al. demonstrated that upon MCMV infection, the SG was able to induce both CD4 and CD8 T_RM_ populations [29], as defined by the expression of CD11a, CD69, and CD103 on T cells [60]. These T_RM_ populations can confer immediate protection against MCMV re-infections. They are mostly excluded from the circulation, and whereas CD8 T_RM_ induction is independent of local cognate antigen presence, CD4 T cell tissue maintenance and T_RM_ formation is strictly antigen-dependent. However, surprisingly and contrary to primary infection, where CD4 T cells are the only cells capable of controlling the infection, CD8 T_RM_ cells seem to be superior in contributing to local protection compared to memory CD4 T cells in the SG [29]. Nevertheless, this study highlighted the capacity of M25-specific CD4 T cells to form and maintain T_RM_ cells.

Another axis of CD4 T cell-mediated control which has not been extensively explored is how they coordinate the MCMV-specific ab response. Indeed, the SG at a steady state is known to harbor a CD4 T cell population producing high levels of IL-5, a cytokine required for IgA production [15,61]. Although early studies showed that antibodies were not essential for the resolution of primary MCMV infection, they were shown to limit the dissemination of recurrent infections [62]. There is also little information about the presence or role of CD4 T follicular helper cells (Tfh) in the MCMV-infected SG, but a few studies suggest an involvement in the formation of ectopic germinal centers in MCMV-infected SGs [63], or the presence of Tfh in the context of Sjögren’s syndrome [64,65]. Of note, SGEC can produce IL-6 and express ICOS-L, two factors important for the differentiation of Tfh [51,66]. However, additional studies are needed to confirm the presence or role for Tfh in MCMV-infected SGs. 

Finally, and in striking contrast to other organs, CD4 Tregs are critical to prevent viral reactivation in the SGs. Almanan et al. showed that this was achieved by suppressing CD4 Foxp3^−^ IL10^+^ cells, as IL-10 supports MCMV replication in the SGs [16,25]. However, more work is necessary to identify the cellular targets of IL-10 leading to viral reactivation at these later stages of infection. 

Studies on CD4 T cell-mediated control during latency or to prevent re-infection are limited. However, a few studies suggest an underappreciated role at later stages. M25-specific CD4 T_RM_ cells have been shown to confer protection [29], but work on other MCMV-specific populations is missing. Additionally, how CD4 T cells regulate the MCMV-specific antibody response, along with further investigations into CD4 Treg-mediated regulation might complement our understanding CD4 T cell responses during latency in the SG. 

## 6. Conclusions and Outlook

This review summarizes the current understanding of the CD4 T cell-mediated immune control in MCMV-infected SGs. Multiple reports agreed on the Th1 nature of this control, with early studies emphasizing the central role of IFNγ-producing CD4 T cells [21,24,48]. Accordingly, investigations on Th1-related chemokine receptors revealed an organ-dependent function of CXCR3 and CCR5 for the organ infiltration of T cells, with no apparent role for infiltration of the SGs [24,37]. Once infiltrated in the SG and guided by CXCR3 expression towards regions of high CXCL9/10 abundance, CD4 T cells position themselves in clusters around the APCs that have taken up viral cargo. These APCs with engulfed remnants of MCMV-infected cells are inducing TCR signalling in MCMV-specific CD4 T cells, evidenced by Nur77 and IFNγ expression. A short-range of IFNγ diffusion is then proposed to offer local protection [24]. The multiplication of locally IFNγ-protected areas is thought to eventually result in organ-wide protection (Figure 3). Whereas the focus on the Th1 response allowed a significant improvement of our appreciation of the IFNγ^+^ CD4 T cell-mediated control in the MCMV-infected SG, chemokine ligand/receptor pairs other than CXCL9/10–CXCR3 might be involved [43,44].

Interestingly, a growing body of evidence suggests an important heterogeneity of MCMV-specific CD4 T cells with different cytokine profiles and kinetics across several organs, even though for most targeted epitopes, similar kinetics are observed [23,24,27,36,47]. As opposed to most MCMV-specific CD4 T cell responses, the m09-specific CD4 T cell response shows delayed kinetics, raising the question about the reason for this differential kinetics. Finally, the distribution of epitope specificities within CD4 T cells (i.e., CD4 memory T cells) at later stages in the SG remains vastly unknown.

The delayed control by IFNγ^+^ CD4 T cells in the SG has been postulated to be partly due to an indirect antigen recognition on APCs [24]. Another explanation is the high percentage of IL-10 production by FoxP3^−^ CD4 T cells [25,36], suggested to be induced through ICOS signalling [36]. The identification of the exact cellular source of ICOS-L could further help to understand IL-10 production in the SG, possibly leading to a delay in viral control. 

Finally, there seems to be a different mode of regulation of the CD4 T cell response in the SG during active viral replication and latency phases by TRAIL^+^ ILC1’s and FoxP3^+^ CD4 Tregs, respectively [16,58,59]. Further work on these regulatory mechanisms might shed further light on their physiological role, such as the avoidance of immunopathology.

Here, we highlighted the importance of the Th1 response during MCMV infection and the mechanisms by which CD4 T cell-mediated control is achieved, but also the heterogeneity of the MCMV-specific CD4 T cell response in the SG. We propose that extensive longitudinal investigations of the MCMV-specific CD4 T cell populations, coupled with a better understanding of the particular SG tissue immunity, might further advance our understanding about the delayed immune control mediated by CD4 T cells in this tissue.

## Figures and Tables

**Figure 1 pathogens-10-01531-f001:**
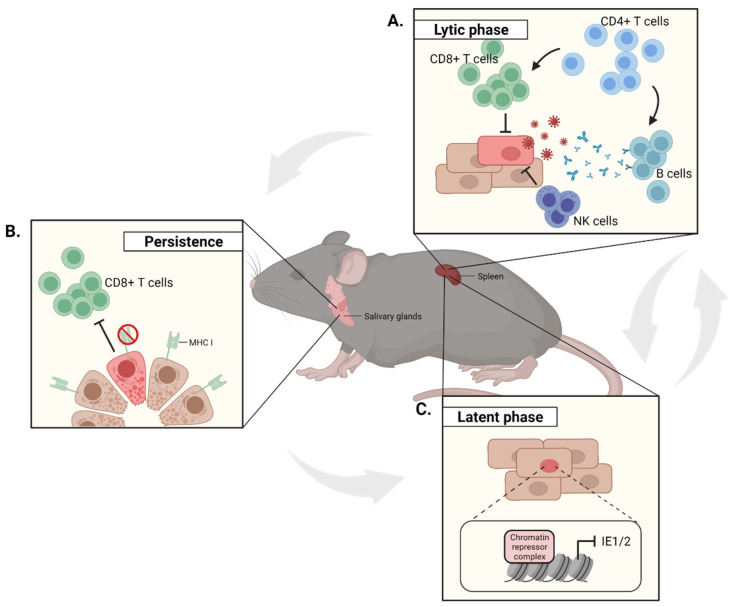
MCMV life cycle. Red cells represent lytically MCMV-infected cells. (**A**) During primary infection, MCMV undergoes lytic replication and disseminates throughout the body. This generates a robust immune response, including neutralizing antibodies, natural killer cells, and T cells, which eventually leads to the control of viral lytic replication and resolution of the primary infection. (**B**) In the salivary glands, immune evasion mechanisms, such as MHC class I downregulation on infected glandular epithelial cells, allow for a persistent lytic replication over several weeks. (**C**) Whereas host antiviral responses are able to resolve the lytic replication, MCMV establishes a multisite latency, such as in the spleen. Through a chromatin repressor complex, the latent transcription program interrupts the production of infectious virions. Reactivation events sporadically occur from these sites and productive replication is rapidly controlled by potent effector-like immune responses.

**Figure 2 pathogens-10-01531-f002:**
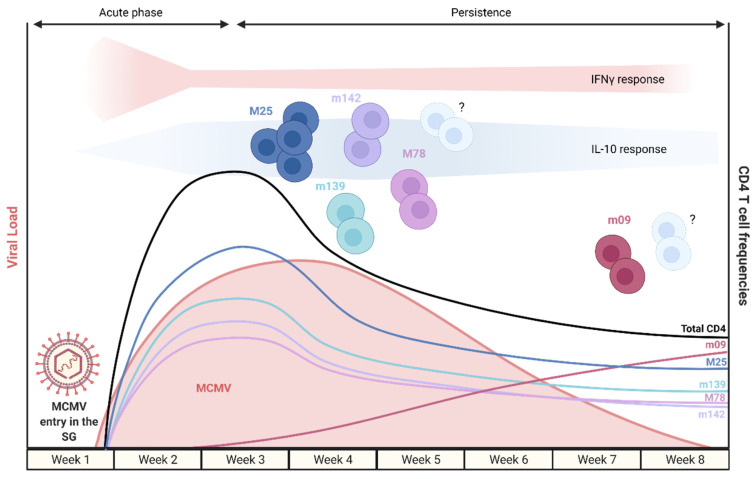
Schematic representation of the virus titer and MCMV-specific CD4 T cell kinetics, along with their relative expression of IFNγ and IL-10, during acute and persistent infection of the SG. The first week of infection in the SG is dominated by an IFNγ response from MCMV-specific CD4 T cells, which is then substituted by an IL-10 response over several weeks, followed by an increase again of the IFNγ response. Of note, the virus load is based on MCMV-3DR kinetics and other MCMV strains might display slightly different kinetics.

**Figure 3 pathogens-10-01531-f003:**
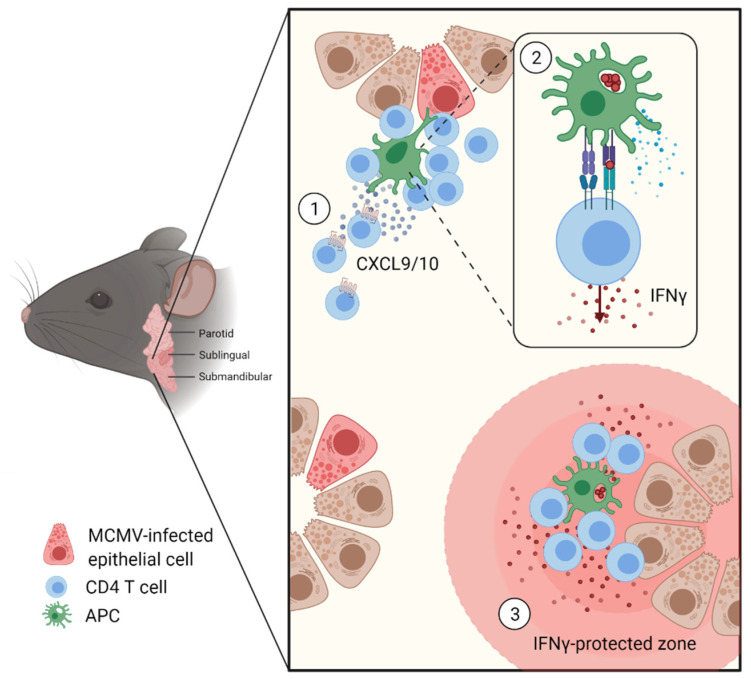
IFNγ^+^ CD4 T cell-mediated immune control in the SGs. (**1**) Guided by CXCL9/10 gradient, CD4 T cells accumulate around APCs with engulfed viral material. (**2**) Within these clusters, IFNγ production is triggered through TCR engagement by APCs. (**3**) The gradient of IFNγ produced by CD4 T cells creates a local IFNγ-protected zone.

**Table 1 pathogens-10-01531-t001:** Reported MCMV peptide specific CD4 T cell populations in the SG.

MCMV-Specific Populations	Sequence	Protein Family/Function	HCMV Homologue	Reported Cytokine Profile	Features	Ref.
m09_133–147_	GYLYIYPSAGNSFDL	Glycoprotein family	−	IFNγ, IL-10	Expand at later time point	[36]
M25_409–423_	NHLYETPISATAMVI	Tegument protein	UL25	IFNγ, IL-10	-	[36]
M25_411–425_	LYETPISATAMVIDI	Tegument protein	UL25	IFNγ, TNFα	Immunodominant epitope	[24,29]
M78_417–431_	SQQKMTSLPMSVFYS	UL78 family/transmembrane receptor homologue	UL78	−	Show cytolytic phenotype in spleen but not in SG	[27]
m139_560–574_	TRPYRYPRVCDASLS	US22 family homologue	−	IFNγ, IL-10	−	[36]
m142_24–38_	RSRYLTAAAVTAVLQ	US22 family homologue	−	IFNγ, IL-10	−	[36]

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
