# Peer review of "CD4 T Cell-Mediated Immune Control of Cytomegalovirus Infection in Murine Salivary Glands"

_pathogens, 2021, doi:10.3390/pathogens10121531_

Round 1

Reviewer 1 Report

This is a very well written article detailing the niche of CMV infection in the SG and the immune surveillance associated with it. The work is very comprehensive and provides excellent insights into the observations made by the group and others.

A minor comment includes the addition of a small section in the first part of the article where the authors could introduce a figure and small summary of the CMV lifecycle. This might provide greater readership to this article. 

Author Response

This is a very well written article detailing the niche of CMV infection in the SG and the immune surveillance associated with it. The work is very comprehensive and provides excellent insights into the observations made by the group and others.

We would like to thank the reviewer for his / her appreciation of the work,

A minor comment includes the addition of a small section in the first part of the article where the authors could introduce a figure and small summary of the CMV lifecycle. This might provide greater readership to this article. 

We added a figure and a section on the CMV lifecycle in the introduction.

Reviewer 2 Report

The manuscript by Zanger et al reviews the role of CD4+ T cells in the salivary glands of MCMV-infected mice.  The salivary gland plays a key role in MCMV natural history in which MCMV replication continues past when the virus is replicating in other organs.  This is an excellent summary of what is known about the role of CD4 cells in controlling MCMV in the salivary gland, as well as key aspects that require further investigation.  It effectively highlights that CD4 cells in the salivary gland operate exhibit a different profile suite distinction from MCMV-specific CD4 T cells in other organs.  Presumably, the different functional activities of cd4 T cells in the salivary gland is due to some virus-mediated effect to enable MCMV to replicate in and be shed from the salivary glands to enable transmission to susceptible cohorts.  The review is excellent as is, but one suggestion for highlighting these findings to the human condition would be to briefly include the role of the salivary gland in enabling horizontal transmission of HCMV in saliva.

Author Response

The manuscript by Zanger et al reviews the role of CD4+ T cells in the salivary glands of MCMV-infected mice.  The salivary gland plays a key role in MCMV natural history in which MCMV replication continues past when the virus is replicating in other organs.  This is an excellent summary of what is known about the role of CD4 cells in controlling MCMV in the salivary gland, as well as key aspects that require further investigation.  It effectively highlights that CD4 cells in the salivary gland operate exhibit a different profile suite distinction from MCMV-specific CD4 T cells in other organs.  Presumably, the different functional activities of cd4 T cells in the salivary gland is due to some virus-mediated effect to enable MCMV to replicate in and be shed from the salivary glands to enable transmission to susceptible cohorts.  The review is excellent as is, but one suggestion for highlighting these findings to the human condition would be to briefly include the role of the salivary gland in enabling horizontal transmission of HCMV in saliva.

We would like to thank the reviewer for the appreciation of your work. We mention in the introduction that “The salivary glands (SG) are a preferred mucosal site of persistent CMV replication, playing a central role in horizontal transmission in both humans and mice.”

Reviewer 3 Report

The review article "CD4 T Cell-Mediated Immune Control of Cytomegalovirus  Infection in Murine Salivary Glands" by Zangger et al describes CD4 T cell-mediated responses in salivary glands and how various immune components influence the CD4 T cells' action. The article is written clearly and covers most of the available literature related to MCMV. However, a few things should be addressed/rewritten to make it more clear.

  1. Each section is not well-separated by the information otherwise the title could be rewritten relevant to the content that comes under. The current format makes readers go back to different sections to see what related information we read there. For eg, the introduction has information about CD4 T cells /cytokines similar to the "phenotype of MCMV specific cells in SG" section, also CD4 T cell dynamics section starts with an effector-like phenotype and describes mostly chemokine responses. I would rewrite the title as for eg chemokine driven CD4 T cell changes to be relevant to the content.
  2. Fig 1 - the legend is not very clear. The shaded red curve is MCMV load and different curve lines are CD4 T cell freq with different specificities? if yes, please give red color to the axis label for viral load, I tend to match with a black line that actually shows total CD4 freq.  Also, the legend says down: Expression of IFNg and  IL-10 throughout MCMV infection. But, they appear on top not down. Cells with unknown specificity are given in the context of IL-10 expression and the other appears at a later week. is this true?
  3. Certain information/keywords are seen in the figure legend but nowhere in the article, I could find the explanation. For eg MCMV-3DR in fig 1 legend; IFN-g protected zone in fig 2 legend, but the article does not talk about its significance
  4. The author could give their insights on delayed control in SG by CD4 T cells is because they are not canonical cytotoxic cells, unlike CD8 T cells that control MCMV in other glands.
  5. Under section 2, I read"MCMV infection, effector CD4 T cells infiltrate the SG..". is it terminally differentiated? are there different phenotypes in mice for terminally differentiated and effector cells?
  6. the author didn't talk about CD4:CD8 T cell ratios, the cytokine IL5 that is found at high levels in SG whether it modulates CD4 T cell response in SG during MCMV infection? 
  7. Epitope specificities are written using the different cases, for eg M78417-431, m139560-574. is there a reason please describe.

Author Response

The review article "CD4 T Cell-Mediated Immune Control of Cytomegalovirus  Infection in Murine Salivary Glands" by Zangger et al describes CD4 T cell-mediated responses in salivary glands and how various immune components influence the CD4 T cells' action. The article is written clearly and covers most of the available literature related to MCMV. However, a few things should be addressed/rewritten to make it more clear.

  1. Each section is not well-separated by the information otherwise the title could be rewritten relevant to the content that comes under. The current format makes readers go back to different sections to see what related information we read there. For eg, the introduction has information about CD4 T cells /cytokines similar to the "phenotype of MCMV specific cells in SG" section, also CD4 T cell dynamics section starts with an effector-like phenotype and describes mostly chemokine responses. I would rewrite the title as for eg chemokine driven CD4 T cell changes to be relevant to the content.

We have accordingly revised the title of the relevant section.

  1. Fig 1 - the legend is not very clear. The shaded red curve is MCMV load and different curve lines are CD4 T cell freq with different specificities? if yes, please give red color to the axis label for viral load, I tend to match with a black line that actually shows total CD4 freq.  Also, the legend says down: Expression of IFNg and  IL-10 throughout MCMV infection. But, they appear on top not down. Cells with unknown specificity are given in the context of IL-10 expression and the other appears at a later week. is this true?

The legend has been clarified and the axis changed. The legend has been developed to help the understanding of the IFNg and IL-10 response by MCMV-specific CD4 T cells throughout MCMV infection.

  1. Certain information/keywords are seen in the figure legend but nowhere in the article, I could find the explanation. For eg MCMV-3DR in fig 1 legend; IFN-g protected zone in fig 2 legend, but the article does not talk about its significance

We now introduced MCMV-3DR, following the references of two articles mentioning it. We also discuss now the role of “IFNg protected areas” in section 4, as possibly resulting in organ-wide protection.

  1. The author could give their insights on delayed control in SG by CD4 T cells is because they are not canonical cytotoxic cells, unlike CD8 T cells that control MCMV in other glands.

We further emphasized on the absence or negligible role of cytotoxic CD4 T cells for the control of lytic infection in the SGs in section 3.

  1. Under section 2, I read "MCMV infection, effector CD4 T cells infiltrate the SG..". is it terminally differentiated? are there different phenotypes in mice for terminally differentiated and effector cells?

Following this comment, we develop that the infiltrating CD4 T cells are CD44hi LFA-1+ CD49d+, indicative of an effector-like phenotype.

  1. the author didn't talk about CD4:CD8 T cell ratios, the cytokine IL5 that is found at high levels in SG whether it modulates CD4 T cell response in SG during MCMV infection? 

We added the information on IL-5, where we suggest that the coordination of the MCMV-specific ab response by CD4 T cells in terms of control needs to be explored further (Section 5). We also emphasized the higher numbers of CD8 T cells compared to CD4 T cells in the SG in the introduction.

  1. Epitope specificities are written using the different cases, for eg M78417-431, m139560-574. is there a reason please describe.

MCMV genes with the uppercase prefix “M” have homologs in HCMV, whereas genes with the lowercase prefix “m” share no sequence identity with HCMV genes. We added this information.